# Time-Course Responses of Apple Leaf Endophytes to the Infection of *Gymnosporangium yamadae*

**DOI:** 10.3390/jof10020128

**Published:** 2024-02-03

**Authors:** Yunfan Li, Siqi Tao, Yingmei Liang

**Affiliations:** 1The Key Laboratory for Silviculture and Conservation of Ministry of Education, Beijing Forestry University, Beijing 100083, China; liyunfan@bjfu.edu.cn (Y.L.);; 2Ecological Observation and Research Station of Heilongjiang Sanjiang Plain Wetlands, National Forestry and Grassland Administration, Shuangyashan 518000, China; 3Museum of Beijing Forestry University, Beijing Forestry University, Beijing 100083, China

**Keywords:** foliar endophytes, plant pathogens, rust infection, plant-microbe interactions

## Abstract

Apple rust, caused by *Gymnosporangium yamadae,* poses a significant challenge to apple production. Prior studies have underscored the pivotal role played by endophytic microbial communities, intimately linked with the host, in influencing plant diseases and their pathogenic outcomes. The objective of this study is to scrutinize alternations in endophytic microbial communities within apple leaves at different stages of apple rust using high-throughput sequencing technology. The findings revealed a discernible pattern characterized by an initial increase and subsequent decrease in the alpha diversity of microbial communities in diseased leaves. A microbial co-occurrence network analysis revealed that the complexity of the bacterial community in diseased leaves diminished initially and then rebounded during the progression of the disease. Additionally, employing the PICRUSt2 platform, this study provided preliminary insights into the functions of microbial communities at specific disease timepoints. During the spermogonial stage, endophytic bacteria particularly exhibited heightened activity in genetic information processing, metabolism, and environmental information processing pathways. Endophytic fungi also significantly enriched a large number of metabolic pathways during the spermogonial stage and aecial stage, exhibiting abnormally active life activities. These findings establish a foundation for comprehending the role of host endophytes in the interaction between pathogens and hosts. Furthermore, they offer valuable insights for the development and exploitation of plant endophytic resources, thereby contributing to enhanced strategies for managing apple rust.

## 1. Introduction

*Gymnosporangium yamadae*, the causative agent of Japanese apple rust, is a heteroecious parasitic and demicyclic fungus that produces teliospores, basidiospores, spermatia, and aeciospores throughout its life cycle. This fungus necessitates two distinct hosts, namely *Malus* spp. and *Juniperus chinensis*, to complete its entire life cycle [1]. When spring rain arrives, teliospores on juniper absorb water and generate basidiospores, which are then dispersed by the wind and eventually settle on apple leaves. Then, spermagonia and aecia are formed on the apple leaves. In autumn, the aeciospores infect the tender branches of junipers again, completing the entire life cycle [2]. The prevalence of apple rust poses a serious threat to the Chinese apple industry [3]. Given the economic significance of apples in the country, this fungal pathogen has resulted in major production and economic losses for fruit farmers in China [4,5]. Consequently, addressing the challenges posed by *G. yamadae* is crucial to safeguarding the development and sustainability of the Chinese apple industry. Currently, research on this disease primarily focuses on pathogens’ classification and chemical control [6,7,8]. The pathogenic mechanism and the process of pathogen–host selection remain poorly understood. Furthermore, *G. yamadae*, like most rust fungi, cannot be cultivated using regular laboratory methods [9], adding further complexity to research endeavors and impeding disease control efforts and follow-up research on pathogens.

Over the past few decades, there has been a paradigm shift in recognizing that the study of plant disease should not solely focus on the interaction between hosts and pathogens but should also consider the role of microorganisms that parasitize plants [10]. This shift in perspective stems from the realization that plants are more accurately seen as complex ecosystems, and disease occurrence should be understood from a holistic, systemic standpoint [11]. Research has shown that endophytes are not passive bystanders but active participants in plant-microbe interactions, greatly impacting the disease development process and the defense capabilities of plants [10]. For example, certain endophytic bacteria found in seeds have been identified to confer disease resistance on the plants, reducing the chances of contracting seed-transmitted diseases [12]. Similarly, entomopathogenic fungi can alter the structure of the host’s endophytic microbial community, enhancing the plant’s resistance against specific diseases such as corn leaf blight [13]. Some endophytes within the host can serve as antigens, triggering the plant’s immune response and inducing plant disease resistance. A study has shown that the pathogen *Xanthomonas oryzae* pv. *oryzae* can stimulate the growth of beneficial bacteria in rice plants, and the presence of a “non-pathogenic pathogen” within the endophytic microbial community accelerates the host’s immune response [14]. While many endophytes play beneficial roles, some can also collaborate with pathogens to facilitate disease occurrence and progression. For example, microbial communities within potato tubers contribute to the degradation of potato tissue by releasing various enzymes, thus aiding in the invasion of *Pectobacterium* species, a pathogen associated with potato disease [15]. Therefore, it is important to study the changes that occur in the host’s endophytic microbiome during the disease process, which is crucial for understanding the mechanisms behind disease occurrence and progression.

Previous research has often focused on specific disease symptom nodes, neglecting to compare the microbial changes throughout the entire disease outbreak [16,17,18]. A few studies have investigated the temporal variations of microbial communities in plant diseases [19,20]. Exploring the changes in microbial groups and functions throughout the entire disease development process can provide a better understanding of the role of the endophytic microbiome in rust disease.

In our previous study, certain microbes were found to be enriched in two sporulation stages of *G. yamadae*-infected leaves compared to healthy leaves [21]. However, this study overlooked the changing patterns of microbial communities as disease processes. Another study analyzed the phyllosphere microbial communities in *G. yamadae*-infected crabapple leaves from field sampling at six developmental stages using amplicon-based methods and revealed that the pattern of microbial communities varied between *Malus* varieties [22]. However, it is important to note that natural diseases in the field exhibit randomness, making it difficult to ensure consistent sources of pathogens within samples, consistent levels of pathogen invasion, and the identification of disease progression stages. These limitations can be overcome by utilizing artificial inoculation, which offers unparalleled advantages for controlled experiments.

In this study, we aimed to investigate the changes in the endophytic microbial community of apple leaves at nine successive stages of apple rust disease after *G. yamadae* basidiospore inoculation. Initial samples were collected based on the pathological characteristics of the host tissues affected by the rust fungus, and subsequently selected samples based on changes in disease symptoms on apple leaves. This series of samplings allows us to observe the progress of the disease course. Through this experiment, we aim to investigate the following scientific questions: (1) How does the endophytic microbial community in apple leaves respond to infection by *G. yamadae* in terms of their structure, diversity, and abundance? (2) What are the potential functions of the endophytic microbial community during disease development? (3) Which specific microorganisms are present during different stages of the disease, and what is their relationship with the disease and their potential role in the pathogenesis process? (4) How does the overall microbial co-occurrence network respond to the invasion by *G. yamadae*?

## 2. Materials and Methods

### 2.1. Site Description and Sampling

The field sampling for this study was conducted at the carefully maintained nursery of the Forest Protection Experimental Station in Haidian District, Beijing (40°0′31″ N, 116°20′26″ E). Three-year-old and one-meter-tall Gala apple seedlings used in the experiment were planted on 20 March 2021. The sampling for this study began at 8:00 on 28 April 2021. The artificial inoculation followed the method described previously [21]. Briefly, the apple leaves in the experimental group were treated with a basidiospore suspension of *G. yamadae* (concentration: 1 × 10^6^ spores/mL) that germinated in sterile water, while the control group was sprayed with sterile water alone. In the experimental group, samples were taken at nine different time points (stages 1–9) (measured by the time of inoculation): 6 h (h), 24 h (h), 3 days (d), 5 days (d), 11 days (d), 14 days (d), 20 days (d), 70 days (d), and 90 days (d). Each time presents a specific stage of the disease’s progression. For example, 6 h marked the time when the rust basidiospores started producing sterigmata. Similarly, 24 h was the period when germ tubes produced by basidiospores first invaded the host’s epidermal stomata under the light microscope. Furthermore, 3 d was the time when the hyphae elongated into a filamentous form and adhered to the cell wall of the host. Other time points also represented different stages, such as the production of haustoria (5 d), the initial spermogonia stage (11 d), the spermogonia maturity stage (20 d), the initial aecial stage (70 d), and the aeciospore mature stage (90 d) (Figure 1 and Appendix A).

During the co-occurrence network analysis, the data were further divided into four phases: A, B, C, and D. These phases were defined as follows: A phase (6 h and 24 h), B phase (3 d and 5 d), C phase (14 d and 20 d), and D phase (70 d and 90 d). The sampling points for the control group were consistent with those of the experimental group. During sampling, apple leaves with the same leaf age were selected; five leaves of the same leaf age were considered as one sample. Three replicates were set up for each sampling point in both the experimental group and the control group. This required a total of 15 leaves for each sampling point in both groups. There were two variables in the experiment, namely whether to inoculate rust fungi and the number of days (or hours) of inoculation.

After collecting the leaves, they were promptly transported to the laboratory for further processing. In the laboratory, the leaves underwent surface sterilization treatment on an ultra-clean workbench. Initially, the leaf surface was disinfected with 1% sodium hypochlorite for 2 min and then washed twice with sterile water for one minute each time. This process ensured the sterility of the leaf surface [23]. Subsequently, surface-sterilized plant tissues were rapidly frozen in liquid nitrogen and then stored in a −80 °C refrigerator for subsequent DNA extraction and sequencing.

### 2.2. DNA Extraction and PCR Amplification

Microbial DNA was extracted from apple leaf samples using the E.Z.N.A.^®^ soil DNA Kit (Omega Bio-Tek, Norcross, GA, USA) according to the manufacturer’s protocols. The final DNA concentration and purification were determined by a NanoDrop 2000 UV-vis spectrophotometer (Thermo Scientific, Wilmington, NC, USA), and DNA quality was checked by 1% agarose gel electrophoresis. Illumina Miseq sequencing was employed to investigate the diversity of leaf endophytic bacterial and fungal communities in healthy and diseased apple leaves. The primer pairs 799F/1193R and ITS1F/ITS1R were used for the amplification of the bacterial 16S rRNA gene and the fungal ITS1 gene, respectively. The PCR primers and conditions are detailed in Appendix A.

### 2.3. Library Preparation for Illumina MiSeq Sequencing

PCR products were recovered using 2% agarose gel and purified using the AxyPrep DNA Gel Extraction Kit (Axygen Biosciences, Union City, CA, USA), then eluted by Tris–HCl, detected by 2% agarose electrophoresis using Quantifluor-ST (Promega, Madison, WI, USA), and quantified according to the Illumina Miseq platform (Illumina, San Diego, CA, USA) Standard Operating Procedures for generating sequencing libraries from the purified amplified fragments.

### 2.4. Bioinformatics Analysis

The analysis was conducted by following the “Atacama soil microbiome tutorial” of QIIME2 documents along with customized program scripts (https://docs.qiime2.org/2019.1/, accessed on 1 January 2019). Briefly, raw data FASTQ files were imported into the format that could be operated by the QIIME2 system using the ‘qiime tools import’ program. Demultiplexed sequences from each sample were quality filtered and trimmed, denoised, and merged, and then the chimeric sequences were identified and removed using the QIIME2 dada2 plugin to obtain the feature table of amplicon sequence variant (ASV) [24,25], excluding the sequences of *G. yamadae* that we inoculated. Bacteria and fungi were classified using the SILVA and UNITE reference databases, respectively. Contaminating mitochondrial and chloroplast sequences were filtered using the QIIME2 feature-table plugin. Appropriate statistical methods, including ANOVA, Kruskal–Wallis, LEfSe, and DEseq2, were employed to identify the bacteria and fungi with different abundances among samples and groups [26,27,28]. Diversity metrics were calculated using the core-diversity plugin within QIIME2. Feature-level alpha diversity indices, such as observed ASVs, the Chao1 richness estimator, and the Shannon diversity index, were calculated to estimate the microbial diversity within an individual sample. Beta diversity distance measurements, including Bray Curtis, unweighted UniFrac, and weighted UniFrac, were performed to investigate the structural variation of microbial communities across samples and then visualized via principal coordinate analysis (PCoA) and nonmetric multidimensional scaling (NMDS) [29]. Partial least squares discriminant analysis (PLS-DA) was also introduced as a supervised model to reveal the microbiota variation among groups. Co-occurrence analysis was performed by calculating Spearman’s rank correlations between predominant taxa and the network plot, used to display the associations among taxa, using the Gephi 0.10.0 version for network visualization analysis [30]. In addition, the potential KEGG Ortholog (KO) [31] functional profiles of microbial communities were predicted with PICRUSt2-2.4.2 [32] and annotated by the MetaCyc database [33].

## 3. Results

### 3.1. A Series of Symptoms Change during Apple Rust Infection

During the occurrence and development of rust disease, various symptoms manifest in apple leaves, reflecting the progression of the disease. This study meticulously tracked the timelines of symptom development post-artificial inoculation, shedding light on distinct stages of rust disease. The disease evolves over approximately 3 months, from the initial microscopic changes to the eventual blight of the whole leaf. The key stages of the disease progression are as follows: At 6 h, spore germination becomes observable under an optical microscope; by 24 h, fungal hyphae are evident, invading the host stomata. Starting from the third day onwards, faded spots (an indicator of mass haustoria production) emerge on the leaves; at 5 d, the color of these faded spots transitions to orange. At 11 d, multiple orange spots coalesce into patches, and needle-like black spots appear on the spots at 14 d. Around 20 d, large drops of nectar are produced around the orange spots, and aecia emerge from swollen, thickened plant tissues at 70 d. Finally, at 90 d, the aecia attains maturity, releasing numerous aeciospores from the aecia (Figure 1 and Appendix A).

### 3.2. Composition of Bacterial and Fungal Communities in the Endophytes of Apple Leaves

The amplicon sequencing yielded a total of 2,793,228 bacterial sequences and 2,655,942 fungal sequences. Rarefaction curves reached a plateau, indicating that the sequencing depth was sufficient to capture the species diversity of all samples (Appendix A). In both healthy and diseased leaf samples, bacterial sequences were clustered into 5592 and 6461 amplicon sequence variants (ASVs), respectively, with 1901 common OUTs. As for endophytic fungi, the sequences were grouped into 2159 and 2247 ASVs in healthy and diseased leaf samples, respectively, with 283 shared ASVs between the two groups. The endophytic bacteria ASVs were predominantly annotated at the genus level, while fungal ASVs were identified at the phylum level (Appendix A), encompassing 37 phyla, 98 classes, 160 orders, 261 families, and 551 genera for bacteria, and 4 phyla, 12 classes, 26 orders, 42 families, and 43 genera for fungi (Appendix A). The dominant phyla in the endophytic bacterial community of apple leaves were Proteobacteria, Firmicutes, Actinobacteria, Bacteroidetes, and Acidobacteria. Proteobacteria constituted the largest proportion, approximately 72% of the total. At the class level, Betaproteobacteria, Gammaproteobacteria, Alphaproteobacteria, Bacilli, Actinobacteria, and Clostridia were the major groups, with Betaproteobacteria being the predominant class, accounting for 32% of the total. The endophytic fungal community was relatively simple and primarily composed of Ascomycota and Basidiomycota at the phylum level, with Ascomycota accounting for more than 60% of the total. Dominant classes included Dothideomycetes, Eurotiomycetes, Sordariomycetes, Agaricomycetes, and Pezizomycetes (Appendix A). A smaller proportion of bacteria and fungi were categorized in the “other group” (Appendix A).

### 3.3. Changes in Endophytic Bacterial and Fungal Communities during Disease Progression in Apple Leaves

Principal Coordinate Analysis (PCoA) and Permutational Multivariate Analysis of Variance (PERMANOVA), based on the weighted UniFrac distances of bacterial and fungal communities, elucidated that the changes in microbial communities were mainly explained by the different stages of the disease, which revealed significant differences in the composition of bacterial and fungal communities as the disease progressed gradually (Figure 2). A more in-depth comparison of microbial composition structure among different disease phases was compared using PERMANOVA analysis, which indicated significant differences (*p* < 0.05) between all phases except for the fungal communities between phases A and B. However, only half of the healthy samples showed significant differences (Appendix A).

Differences in alpha diversity indices determined by the Shannon index were analyzed for both bacteria and fungi (Appendix A). The alpha diversity analysis results showed that, in general, the alpha diversity of bacterial communities was higher than that of fungal communities. The results demonstrated a pattern of initially increasing and subsequently decreasing alpha diversity in microbial communities. Overall, diseased leaves exhibited lower bacterial diversity compared to healthy leaves in the first half of the disease, yet showed higher values compared to the healthy group in the latter half of the disease. In contrast, fungal diversity surpasses that of healthy leaves during the whole disease. Conversely, the control group remained relatively stable.

The number of ASVs in each stage was also analyzed. Before stage 6 (14 d), the number of bacterial ASVs was higher in the healthy group compared to the disease group. However, at stages 6 (14 d) and 7 (20 d), the number of ASVs in the disease group dramatically increased, surpassing that of the healthy group. As for fungi, the number of ASVs in the disease group started to exceed those in the healthy group from stage 3 onwards, peaking at stage 7 (Appendix A).

Throughout the whole course of disease progression, at the phylum level, Proteobacteria maintained a high abundance before stage 6, but experienced a sharp decrease at stage 6, followed by a slight increase that did not return to the previous high levels. Conversely, Firmicutes, Actinobacteriota, and Bacteroidota initially had low levels but saw a substantial increase after stage 6. At the class level, Betaproteobacteria accounted for over 60% of the relative abundance, starting to decrease from stage 2, while Gammaproteobacteria started to increase from stage 2, experienced a sharp decrease at stage 6, and maintained a low abundance thereafter. Similarly, Bacilli also experienced a shift at stage 6, maintaining a low abundance before and a significant increase after. In terms of endophytic fungi, the majority of them belonged to Ascomycota. At the class level, Eurotiomycetes showed a consistent decrease from stage 1 to stage 4, followed by higher abundance at stages 5, 6, and 7. However, Eurotiomycetes was completely replaced by Dothideomycetes at stages 8 and 9, with a trend opposite to that of Eurotiomycetes (the information in this paragraph is derived from Figure 3a). Community abundance at the taxonomic level is shown in the accompanying Appendix A.

According to a specific sampling time point, compared to the healthy control group, the diseased group showed a great difference. At stage 2 (24 hpi), when basidiospores germinated and entered the leaf stomata, there were significant changes in the endophytic bacteria. At the phylum level, Proteobacteria increased in relative abundance, while Firmicutes, Actinobacteria, and Bacteroidota decreased (Figure 3a). At the genus level, the abundance of *Pseudomonas* and *Brevundimonas* decreased, but the abundance of *Ralstonia* and *Delftia* increased (Figure 3b). For the endophytic fungal community, Basidiomycota increased and Ascomycota decreased; Pucciniomycetes and Dothideomycetes increased in abundance, while Eurotiomycetes and Sordariomycetes decreased. The relative abundances of *Penicillium* and *Discosia* showed significant changes, with one increasing and the other decreasing (Figure 3b). During the spermogonium stage (stages 3–7), there were few relative fluctuations in the relative content of microflora. At the maturity of the spermogonia (stages 5–7), Firmicutes, Actinobacteriota, and Bacteroidota all increased remarkably, while Proteobacteria decreased significantly. In the aecidium stage, Proteobacteria regained dominance in the bacterial community. For endophytic fungi, the relative content of Ascomycota was lower than that of healthy samples at stage 2 and stage 3, but there were no significant changes at other sampling points (Figure 3a).

Through LEfSe analysis, specific microorganisms’ relative abundance at each stage of the disease was identified (Figure 3c, Appendix A). In terms of endophytic bacteria, during the germ tubes (stage 1), *Ralstonia*, *Proteiniphilum*, and *Methyloversatilis* were the specific microorganisms. During the stage when the mycelium successfully infected the host (stage 3), *Paenibacillus* and *Lysinibacillus* were prominent. The spermogonial stage (stages 6 and 7) was characterized by the presence of *Enterobacter*, *Buchnera*, *Serratia*, *Janthinobacterium*, *Erwinia*, *Acholeplasma*, Acholeplasmatales, *Clostridium*, *Streptococcus*, *Bradyrhizobium*, *Mycobacterium*, *Ammoniphilus*, *Variovorax*, *Neisseria*, *Macellibacteroides*, *Pedomicrobium*, *Ruminococcus*, *Rothia*, *Glycomyces*, *Anaerovorax*, LCP6, *Cloacibacterium*, *Pelotomaculum*, *Corynebacterium*, *Aurantimonas*, *Porphyromonas*, *Propionibacterium*, *Acidovorax*, *Jonesia*, *Thermomonas*, *Brachybacterium*, and *Mycoplana*. The aecial stage (stages 8 and 9) was associated with *Staphylococcus*, *Kineococcus*, *Burkholderia*, Rhodospirillaceae, Xanthomonadaceae, Dermatophilaceae, *Salinispora*, *Dyella*, *Prevotella*, *Enhydrobacter*, *Piscicoccus*, *Tolumonas*, *Anaerococcus*, Alphaproteobacteria, Actinobacteria, *Sphingomonas*, *Telluria*, *Quadrisphaera*, *Roseomonas*, *Methylobacterium*, *Belnapia*, *Bdellovibrio*, and Bdellovibrionaceae. Regarding endophytic fungi, Hypocreales, Filobasiaceae, and Tremellomycetes showed high abundance when the rust mycelia initially colonized the host tissue, while *Filobasidium* was dominant during the spermogonial stage, and Dothideomycetes and *Alternaria* prevailed during the aecidium stage.

### 3.4. Microbial Correlation of Diseased and Healthy Samples

To explore the influence of disease progression on endophytic community interaction, a microbial network interaction analysis was performed on samples collected in four phases (Figure 4). In the bacterial community, several indices (average degree, clustering coefficient, edges, and network density) in the diseased group were generally higher than those in the healthy group, except for modulization and nodes being higher in the healthy samples. The fluctuations observed in the healthy samples were relatively mild across each phase. Specifically, the average degree, edges, network density, and nodes exhibited a declining trend overall. On the contrary, the clustering coefficient and modulization generally increased. Notably, in stage C, the edges, modularity, and nodes reached their highest levels, while the network density reached its lowest level (Appendix A). As for fungi, the diseased group exhibited a higher clustering coefficient and network density compared to the healthy group, similar to the bacterial trends. However, the average degree, edge, modulization, and nodes were lower in the diseased group (Appendix A). The trend observed in the healthy samples showed a general increase, followed by a decrease, and then another increase. In contrast, the diseased samples showed a decrease followed by an increase, with the lowest point reached in stage C.

### 3.5. Microbial Functional Allocation at Different Stages of the Disease Progression in Apple Leaves

The bacterial and fungal sequences were annotated using the KEGG and MetaCyc databases, respectively, to obtain the functional annotation results. PCA results indicated significant differences in the functional profiles of endophytic flora communities during the four diseased phases that were functionally significantly different (Appendix A). When examining each stage of the disease course for endophytic bacteria, metabolism-related pathways were more prevalent in stage A, while stage B was characterized by the enrichment of pathways associated with cellular processes and information processing. In stage C, pathways related to genetic information processing and organismal systems were notably enriched (Figure 5a). In addition, the specific proportions of metabolic pathways enriched by bacteria in different stages of diseased leaves were also shown in Figure 5b,c. On the other hand, the fungal situation was shown in Figure 5d, in which various metabolic pathways are mainly enriched in stages C and D (Appendix A).

## 4. Discussion

In the analysis of endophytic bacterial and fungal communities of healthy and *G. yamadae*-infected apple leaves at different time points, a series of changes were observed in the latter, while the healthy group remained relatively stable. The diversity of endophytic flora in diseased leaves increased, as evidenced by the rise in the number of ASVs and community evenness. Throughout the disease progression, a substantial enrichment of potential pathogens occurred, adversely effecting host growth, consistent with findings in previous studies. Network co-occurrence analysis revealed a decrease in the modularity of the endophytic bacterial community in diseased leaves, indicating reduced community stability. Functional predictions indicated enhanced representative activities, such as cell activity and information exchange in the bacterial community of the diseased leaves. Similar metabolic activities were also observed in the fungal community. Our research provides compelling evidence that the pathogen invasion disrupts the microbial community’s diversity, aggregation, and networks, significantly impacting their ecological functions. Subsequent sections explore the implications of these findings, enhancing our understanding of disease-induced alterations in the assembly, co-occurrence, and functionality of plant microbiomes.

### 4.1. Diversity and Structure Dynamics during Disease Progression

The diversity of bacteria and fungi in healthy samples revealed a stable community with minimal fluctuations in each phase. Conversely, in diseased samples, bacterial and fungal diversity initially showed subtle changes and then rapidly increased after the maturation of spermogonia. This pattern was reflected in both the Shannon index and the abundance of Amplicon Sequence Variants (ASV), with a more pronounced increase in bacterial diversity during the later stages. This stage may represent a critical period of balance between rust pathogens and host immunity. The host’s immune system continuously combated external rust pathogens during the early stages, resulting in minimal changes in diversity. However, as rust pathogens continue to colonize, the host’s cellular structure [34] and its immune system become severely compromised [35]. The balance is broken, leading to a massive influx of foreign communities into the host, thereby triggering a rapid increase in ASV diversity. Definitely, this does not rule out the possibility that some of these microorganisms were “foreign reinforcements” recruited by the host to combat rust pathogens [36]. Simultaneously, the endophytic microbial community strived to resist the invasion by rapidly proliferating, which may be responsible for the observed results. This consistent pattern suggests ongoing host regulations of the endophytic community to adapt to the evolving disease process, emphasizing the dynamic nature of the host-microbe interaction [17]. The observed changes in bacterial and fungal diversity during the later stages indicate a dominance of bacteria in the ecological niche towards the end of the disease course. This observation aligns with the findings from a study on the microbial communities during *G. yamadae* infection in Malus species [19]. The PCoA results for bacteria indicated slight differences in the healthy leaves at each stage, while the communities of diseased leaves significantly exhibited variations at each stage, indicating that the assembly of bacterial communities was profoundly affected by the rust disease. On the other hand, excluding phase D, the fungal community showed significant dispersion from the other stages, while the differences among the remaining stages were not as substantial. This observation suggests that the fungal community was primarily influenced by the development stage, with less impact from the rust infection itself [37]. In addition, the R square value of each stage was greater than 0, and the *p*-value was less than 0.05, which confirms that the differences between sample groups were significantly larger than the differences within groups.

### 4.2. Shift in Composition of Leaves Endophytic Microbial Communities with G. yamadae Infection

Though there were differences in the relative abundance of endophytes between healthy and diseased leaves at various stages of disease development, the composition of high-abundance microorganisms was similar. For bacteria, Proteobacteria, Firmicutes, Actinobacteriota, Bacteroidota, and Acidobacteriota accounted for a high proportion, while the highly abundant taxa in the fungal community were mainly Ascomycota and Basidiomycota, which was consistent with many studies on endophytes in leaf microbiota [13,14,38].

Many members of the Bacilli and Actinobacteria groups have exerted growth promotion and antibacterial effects [39,40]. In this study, the two endophytic bacterial communities were at lower levels in the diseased group before stage 6, which may be a strategy for rust fungi to facilitate their invasion. Previous studies show pathogens could preoccupy nutrients [41], secrete metabolites [42], creating an environment conducive to infection, all for competing with beneficial microorganisms within the host, and even studies showed pathogens inject effector proteins into antagonistic microorganisms as potent toxins to inhibit the growth of microbial competitors, which contribute to establishing microbial communities suitable for pathogens’ invasion in the hosts’ environments [43,44]. The phenomenon of a sharp increase in abundance during stage 6 may be attributed to the host’s recruitment of external microorganisms, which serves as a counterattack mechanism for plant hosts [36].

Meanwhile, the Pleosporales in the fungal community also exhibited a similar trend of change, decreasing first and then increasing during the infection process of rust fungi. Pleosporalean fungi, reportedly, have significant antifungal properties [45,46]. In summary, the fluctuations in endophyte microbial communities may reflect a pathogen’s invasion strategy.

### 4.3. Small Changes in Host Microbiome Composition Predict Disease Outcomes Earlier Than External Symptom Variation

In this study, there were no obvious symptoms until the appearance of flecks on the leaves 5 days after inoculation. However, significant alternations in the endophytic flora were observed at 6 h, 24 h, and 3 days after inoculation. Previous histopathological studies have found that at 6-hour post-inoculation (hpi), corresponding to the spore germination of rust, bud tubes are produced, and the fungi have not yet invaded the host tissues. We speculate that the observed changes in the microbiota during this time may be attributed to the host’s recognition of rust fungi, leading to the release of metabolites that induce these changes, serving as a preemptive, risk-avoiding approach in advance. Most existing studies have concentrated on the microbial community changes occurring after the host exhibits obvious symptoms, often overlooking the changes preceding the manifestation of symptoms [17,47,48]. Moreover, it has been found that changes in rhizosphere microbiome composition can predict the occurrence of diseases earlier than changes in pathogen density [48]. The successful infection of a pathogen and subsequent presentation of symptoms on the host are the outcomes of its interaction with the host’s microbial community, overcoming both the outer and inner host microbial barriers [49]. Therefore, changes in the microbial community are an important indicator of disease onset. In human diseases, the process of disease can be predicted in advance by detecting microbial changes, which allows for the accelerated development of diagnosis and treatment plans [50]. However, at present, in the field of plants, microbial methods for disease detection are still rare, especially for diseases with a long incubation period and precious medicinal materials, endangered ancient trees, and other hosts. Incorporating microbial approaches would be a favorable method of disease control.

### 4.4. Contrary to the Characteristics of Healthy Leaves with More Beneficial Endophytes, the Harmful Flora Became the Characteristic Flora in the Experimental Group

Among the endophytic bacteria, 13 genera were significantly enriched in the diseased leaves. Notably, several of these genera, including *Erwinia*, *Curtobacterium*, *Pantoea*, and *Enterococcus*, were recognized as potential pathogens. In contrast, healthy samples exhibited a significant enrichment of 21 genera, with some being beneficial bacteria such as *Pseudomonas*, *Bacillus*, *Lactobacillus*, and *Flavobacteria*. While endophytes do not induce diseases in their host, specific conditions can lead to the transformation of endophytes into harmful bacteria [51,52]. Typically, plants remain disease-free when they are healthy. However, aging plants, biological stress, or external pathogen infection can trigger the conversion of endophytic bacteria into pathogenic bacteria, resulting in plant diseases [53]. Among the significantly enriched genera found in the diseased leaves, *Erwinia* spp. are known plant pathogens, causing symptoms like plant necrosis, wilting, leaf spots, and soft rot [54,55]. *Erwinia amylovora* causes fire blight (it infects apples) [56]. Certain species in the genus *Curtobacterium* can cause wilt in plants [57]. *Enterococcus* is a typical pathogenic bacterium that can cause a series of diseases in humans and plants. It often occurs in the body after it has been infected with a disease [15,58]. Except for these, *Rathayibacter* [59] and *Novosphingobium* [60] are also pathogenic bacteria. It is noteworthy that *Erwinia* and *Trabulsiella* were specific genera found in the diseased leaves but were not present in healthy leaves. On the other hand, the healthy samples displayed significant enrichment in 16 out of the 21 microflora genera with beneficial effects. Notably, 9 out of the 10 were associated with beneficial bacteria. For instance, *Brevundimonas* could induce plants to improve their tolerance under stressful environments and promote plant growth [61,62], *Pseudomonas* spp. are well-known plant probiotics [63,64], and *Lactobacillus* is widely used as an antagonistic bacteria in the field of biological control [65]. In addition, other enriched genera, such as *Geobacillus* [66], *Stenotrophomonas* [67], *Bifidobacteria* [68], *Chryseobacterium* [68], *Aeromonas* [69], *Comamonas* [70], *Azomonas* [71], *Klebsiella* [72], *Cellulosimicrobium* [73], *Microbispora* [74], *Kocuria* [75], and *Cellulomonas* [76], have been reported to have antibacterial properties or host growth promotion effects. Concerning endophytic fungi, there was little difference between healthy and diseased leaves, with only *Phoma* significantly clustering in diseased leaves. *Phoma* is a type of pathogenic fungus known to cause severe damage to plants [77]. In summary, after inoculation with rust, the bacterial and fungal groups containing potential pathogenic bacteria were significantly enriched in the leaves, whereas healthy leaves harbored many beneficial bacteria. In other words, the diseased leaves experienced a depletion of their otherwise beneficial flora, which could result in reduced resistance to pathogenic bacteria.

### 4.5. Changes in the Co-Occurrence of the Endophytic Microbial Community with the Enlargement of the Course of Disease

Microbial ecological networks play a crucial role in characterizing the relationships among microorganisms within a community, and recent evidence suggests that the characteristics of ecological networks can influence the community’s response to pathogen invasion [16,22]. Modulization within the network enhances its stability when faced with disturbances [78]. In this study, during the four phases (A, B, C, and D) of bacteria and the three phases (B, C, and D) of fungi, the modulization of healthy leaves was found to be higher compared to diseased leaves, indicating a lower stability in the latter. The invasion of pathogens disrupted the balance of the microbial community but also prompted the possibility that, during this process, the host continuously adjusted its microbiota to better adapt to the disease [42]. Microbial networks characterized by high negative correlations are associated with higher microbial community stability, underscoring the significance of competitive relationships between microbial communities in maintaining host health [78,79,80]. Specifically, during phases A, B, C, and D, the proportion of negative correlations among fungi in healthy leaves was higher than in diseased leaves, indicating internal disarray of the fungal communities. Conversely, the negative correlation of endophytic bacteria within diseased leaves was higher than that of healthy ones, which may be related to the host’s immune regulation. Additionally, it is noteworthy that the ecological niche of pathogens is extremely similar to that of endophytes within the host [81,82]. Even endophytic fungi that inhabit healthy plant tissues and grow asymptomatically may evolve from plant pathogenic fungi into non-pathogenic fungi [83,84]. In this study, the pathogen *G. yamadae* itself is a fungus, and it may be more inclined to occupy the ecological niche of endophytic fungi, leading to the disruption and chaos of the fungal community network [17,85,86].

### 4.6. Endophytes Contend with Gymnosporangium yamadae through a Series of Metabolic Processes

Our study primarily focuses on investigating the functions of endophytic bacteria in the occurrence of disease, specifically examining their involvement in metabolism, cellular processes, genetic information processing, and environmental information processing. Compared to healthy leaves, endophytic bacteria in diseased leaves were more active in various metabolic pathways. In terms of cellular processes, pathways related to cell apoptosis in stage A and related to quorum sensing (QS) in stages A and B are significantly enriched. The observed enhanced communication among endophytic bacteria aligns with QS, a process of intercellular communication where endophytic microorganisms can share cell density information and adjust gene expression accordingly to control the expression of virulence factors in external pathogens [87]. These stages coincide precisely with the periods when rust fungi multiply rapidly and invade host tissues, causing damage. In response to the invasion of rust fungi, endophytic bacteria enhanced their communication and associations with each other. During phase B, pathways associated with biofilm formation were enriched, which was often closely associated with enhanced QS [88]. In terms of environmental information, pathways related to the phosphotransferase system (PTS) were significantly enriched. Bacteria, typically inhabiting dynamic and ever-changing environments, have evolved complex regulatory networks that integrate stimuli and pressures to rapidly adapt to these challenging conditions. Once activated, these signaling pathways quickly regulate basic cellular processes, such as DNA replication, cell division, or cell growth [89]. The involvement of PTS in mediating stress responses indicates that, during this period, endogenous bacteria perceived the presence of pathogens and rapidly regulated basic cellular processes through signaling pathways [90,91]. In the field of genetic information processing, there is a remarkable abundance of pathways of the two-component systems, bacterial secretion systems, and ABC transporters, which may be correlated with QS [92]. This finding suggested that it was a critical period for bacteria to communicate closely with other bacteria and plant hosts. Interestingly, in phase C, the subsequent next time point, a rapid increase in bacterial ASV within the infected leaves was observed. This could possibly be a “cry for help” strategy from the microbiota within the host organism in response to the external environment [36]. At this time, the relative abundance of *Brevundimonas* and *Geobacillus* rapidly increased, and these two types of bacteria have been previously studied for their role in promoting plant growth and responding to biotic and abiotic stress [61,66,93]. For endophytic fungi, most metabolic pathways were significantly active during phase C, but the ASV of fungi in diseased leaves had exceeded that in healthy leaves since phase B, indicating that endophytic fungi in apple leaves were more sensitive to rust fungus. This may be related to the metabolites secreted by the host, promoting the growth and diversity of beneficial bacteria in the body and thus competing for ecological niches with pathogens [94]. The relative abundance of *Talaromyces* significantly increased during phase B. Fungi belonging to this genus have been found to produce metabolites that exhibit antagonistic activity against plant pathogens and promote growth in host plants [95,96].

## 5. Conclusions

Traditionally, the interaction between the host and the pathogen has been predominantly explored from the perspective of pathogen effectors. However, this study expands our understanding of the pathogenesis of apple rust by delving into the dynamics of microbial communities. In essence, this study provides insights into the changes in endophytic bacteria and fungi in apple leaves following infection by *G. yamadae* throughout the course of the disease. The alpha diversity analysis indicates that bacteria undergo more significant changes, whereas the fungal community remains relatively stable. This suggests that bacteria may play a crucial role in the disease process, a hypothesis further supported by functional analysis. It becomes evident that endophytic bacterial communities are affected by the presence of rust fungi, engaging in effective communication with their counterparts in response to fungal infection. These bacteria regulate their own metabolism to resist the fungi and protect their own community across various stages of the disease. Additionally, they act as a protective barrier for the host. The ongoing invasion of rust fungus entails not only a battle with the host’s immune system but also a complex interaction with the microorganisms. Understanding this intricate mechanism holds potential for future application in the prevention and treatment of similar diseases.

## Figures and Tables

**Figure 1 jof-10-00128-f001:**
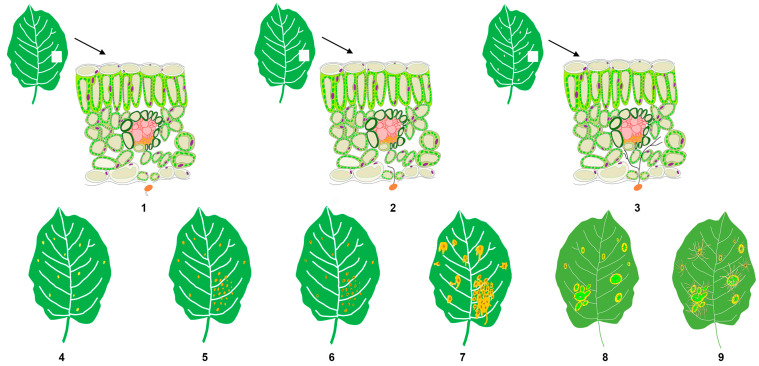
After being infected by *Gymnosporangium yamadae*, apple leaves undergo a series of pathological changes at the micro or macro level (freehand sketching), 1–9 indicates stage 1–9.

**Figure 2 jof-10-00128-f002:**
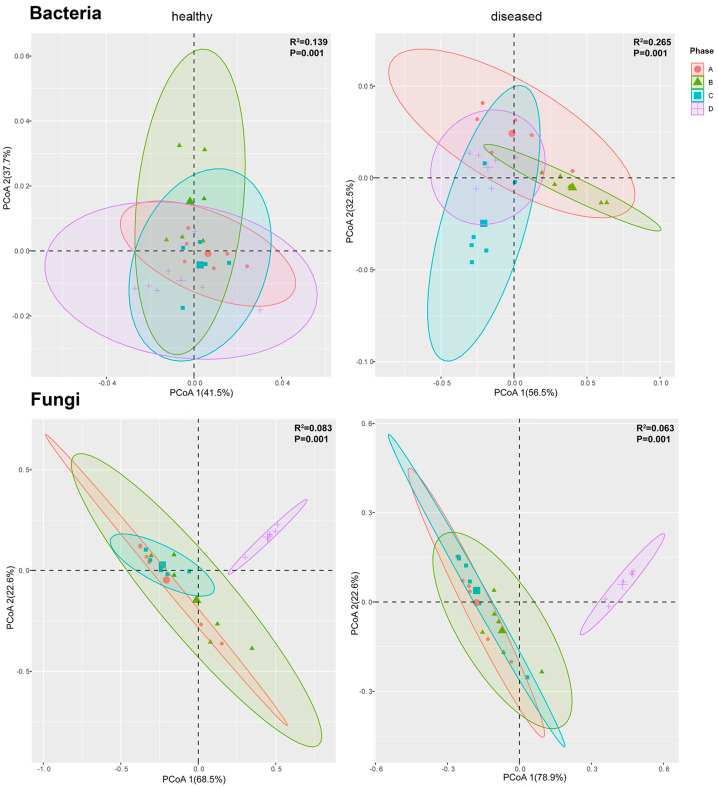
PCoA of bacterial and fungal communities using the weighted UniFrac distance matrix. Samples were sorted according to sample phases (A–D).

**Figure 3 jof-10-00128-f003:**
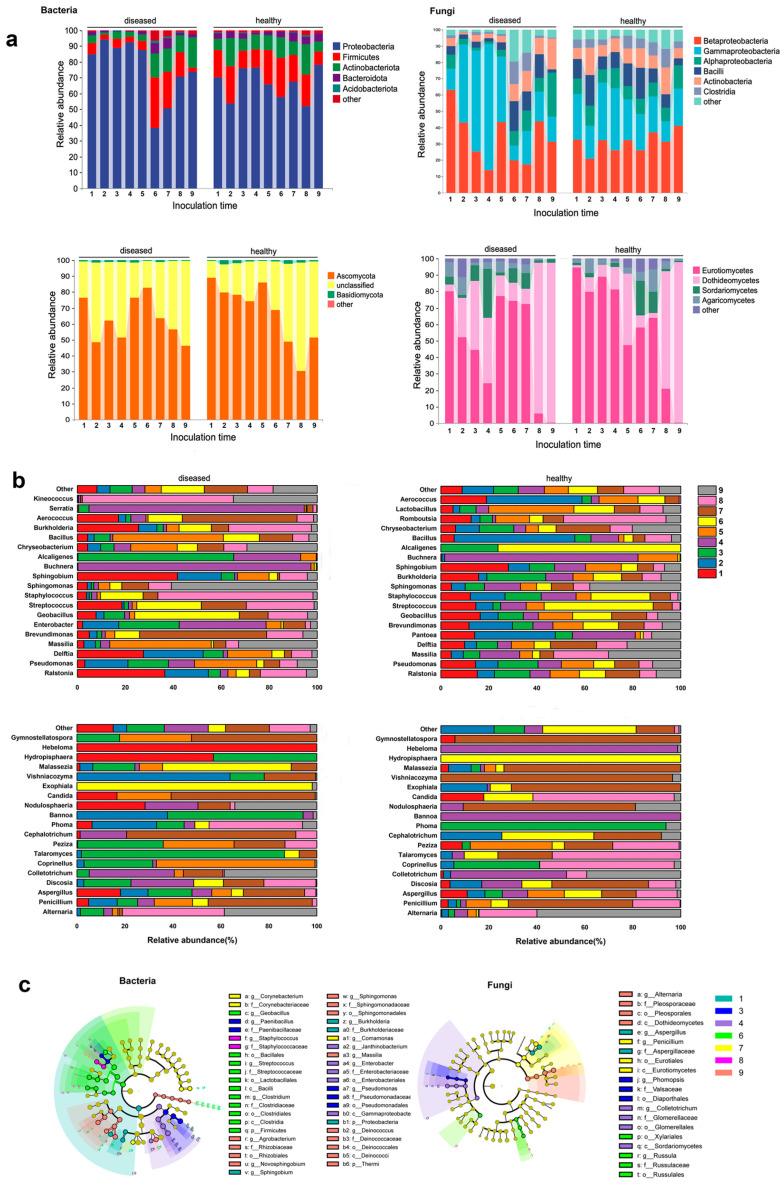
The composition of bacterial and fungal communities in diseased leaves (inoculated with *G. yamadae*) and healthy leaves (not inoculated with *G. yamadae)* at different sampling stages. (**a**) Changes in relative abundance of endophytic bacterial and fungal communities at different sampling stages (at the phylum and class levels). (**b**) At the genus level, the proportion of relative abundances of endophytic bacteria and fungi at each sampling stage. (**c**) Specific endophytic bacterial and fungal communities were significantly enriched at different sampling periods.

**Figure 4 jof-10-00128-f004:**
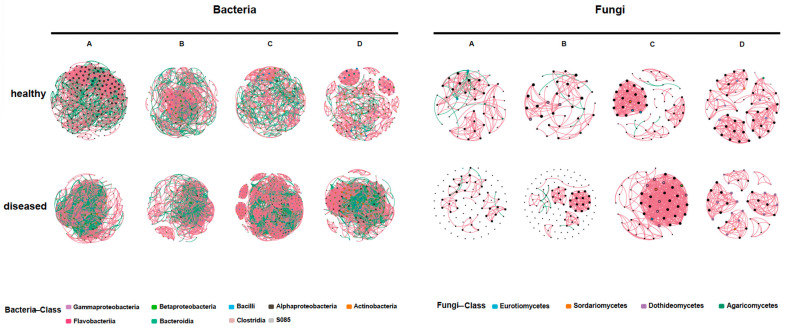
Endophytic microbial co-occurrence networks of noninfected and *G. yamadae*-infected leaves at different sampling stages A, B, C, and D. Positive correlations are shown in red, and negative correlations are displayed in green. The nodes are colored according to the taxonomical classification of microbes (at the class level). The size of the node is proportional to the degree level.

**Figure 5 jof-10-00128-f005:**
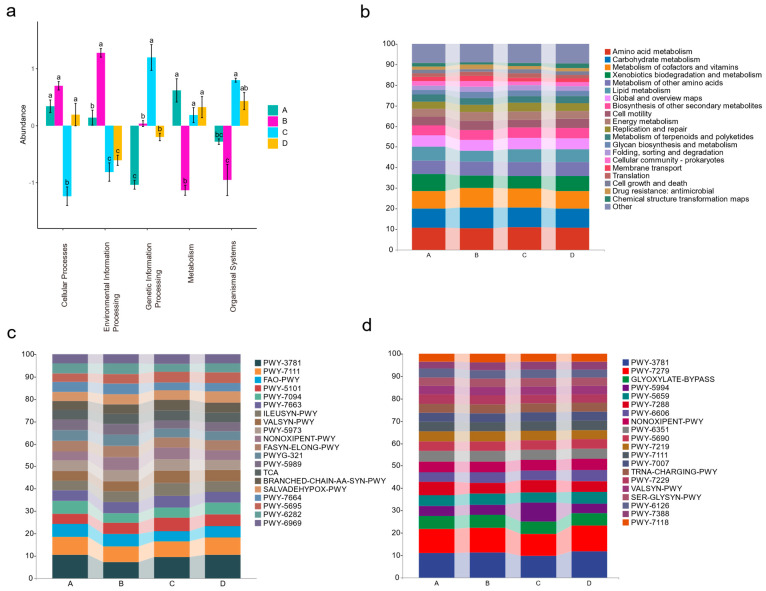
Functional annotation of bacterial and fungal sequences in different sampling phases. (**a**) Functional categories in which bacterial community sequences are significantly enriched at various phases. Different letters indicate statistically significant differences among stages. (**b**) The functional proportion of bacterial communities at each sampling phase was demonstrated in detail; (**a**,**b**) both made use of the KEGG database. Furthermore, (**c**,**d**) showed the functional proportion of endophytic bacterial and fungal communities at different sampling phases (using the MetaCyc database).

## Data Availability

Data are contained within the article and Appendix A.

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
