# Peer review of "Time-Course Responses of Apple Leaf Endophytes to the Infection of Gymnosporangium yamadae"

_jof, 2024, doi:10.3390/jof10020128_

Round 1

Reviewer 1 Report

Comments and Suggestions for Authors

The manuscript titled “Time-course responses of apple leaf endophytes to the infection of Gymnosporangium yamadae” by Yunfan Li et al., Describes changes in endophytic microbial community (in terms of their structure, diversity, and abundance) of apple leaves at nine successive stages in response to G. yamadae infection. The study examines changing pattern and functional profiles of microbial communities as the disease progresses by employing high-throughput sequencing technology, microbial co-occurrence network analysis and the PICRUSt2 platform. Overall, this study provides a baseline understanding of endophytes associated with the apple leaf during G. yamadae infection and analyses of host plant-endophytes-pathogen multipartite complex interactions in the context of pathogenesis and disease management.

There is a possibility for microbial input from fruiting body microbiome (from Juniperus chinensis twig G. yamadae galls as the source of inoculum) even though sterile water was used for the preparation of basidiospores inoculum.

Bacteria Community Inhabiting Heterobasidion Fruiting Body and Associated Wood of Different Decay Classes – Wenzi Ren et al (2022)

https://www.frontiersin.org/articles/10.3389/fmicb.2022.864619/full

Figure 2, 3, and 4 – The resolution of the figures is low, gets blurred when zoomed in.

Comments on the Quality of English Language

Minor typos

Author Response

Dear Reviewer,

Thank you for your time and effort on our manuscript entitled “Time-course responses of apple leaf endophytes to the infection of Gymnosporangium yamadae” (Manuscript ID: jof-2819186). All comments are valuable and very helpful for improving our manuscript. We considered all the comments carefully and highlighted all the changes within the revised manuscript by red Word. In addition, we modified the other parts of the manuscript with the red words. The point-by-point responses to the issues are listed below.

Response: R

Comments 1: There is a possibility for microbial input from fruiting body microbiome (from Juniperus chinensis twig G. yamadae galls as the source of inoculum) even though sterile water was used for the preparation of basidiospores inoculum.

R: It is indeed a problem in our current research. As mentioned in the literature you shared with me, rust fungus itself may have some associated microorganism that cannot be observed under the light microscope. G. yamadae is a demicyclic fungus, which making it very difficult to achieve pure culture. At present, artificial inoculation can only be completed by cultivating teliospores to produce basidiospores. This is a biological limitation of rust, not a technical problem, but the impact on the results of this experiment is very limited. Besides, the symptoms on the apple leaves after the disease in this experiment were consistent with the typical symptoms of the host disease caused by rust bacteria, and the disease could be determined to be caused by rust bacteria, in addition, in the natural state, the rust fungi are also in an impure environment, so the miscellaneous bacteria or other fungi may have little impact on the experimental results of this study. Thank you for this valuable suggestion, which may become a scientific research direction in the future and will be conducive to the understanding of rust research.

Comments 2: Figure 2, 3, and 4 – The resolution of the figures is low, gets blurred when zoomed in.

R: Done. We have replaced with high resolution pictures in revised manuscript, meeting the journal's requirements for manuscript submission (resolution of 300dpi or higher).

Thank you again for your positive and constructive comments and suggestions on our manuscript.

Reviewer 2 Report

Comments and Suggestions for Authors

The article 'Time-course responses of apple leaf endophytes to the infection of Gymnosporangium yamadae', by Yunfan Li, Siqi Tao, Yingmei Liang, depeen different aspects of the interaction between host, pathogen and endophytic microorganisms (bacteria and fungi) in Apple leaf organs. The pathogenetic cycle of the rust agent, the endophytic microbial consortium in healthy and infected leaves and the modifications of this consortium during the various infectious phases of the pathogen are described. This with the aim of providing a contribution to the definition of biological defense strategies against the mycoparasite. The work is well structured, the materials and methods are well described and the results quite innovative and interesting. Only a few small changes are proposed: line 403: infection, instead of infestation; line 480: delete exactly, repeated in next line (or vice versa).

Author Response

Dear Reviewer,

Thank you for your time and effort on our manuscript entitled “Time-course responses of apple leaf endophytes to the infection of Gymnosporangium yamadae” (Manuscript ID: jof-2819186). All comments are valuable and very helpful for improving our manuscript. We considered all the comments carefully and highlighted all the changes within the revised manuscript by red Word. In addition, we modified the other parts of the manuscript with the red words. The point-by-point responses to the issues are listed below.

Response: R

Comments 1: line 403: infection, instead of infestation;

R : Done. Please see line 446.

……, in pathogen density [36]. The successful infection of pathogen and subsequent presentation of symptoms on the host are the……

Comments 2: line 480: delete exactly, repeated in next line (or vice versa).

R : Done. Please see line 525.

……, in various metabolic pathways. In terms of cellular processes, pathways related to cell apoptosis in stage A and related to quorum sensing (QS) in stage A and B are signifi-cantly enriched. The observed enhanced……

Thank you again for your positive and constructive comments and suggestions on our manuscript.

Reviewer 3 Report

Comments and Suggestions for Authors

Dear Colleagues.

There are a few small notes

Line 105….

The field sampling for this study was conducted at the Forest Protection Experi-  mental Station in Haidian District, Beijing (40°0′31″N, 116°20′26″E). Gala apple seedlings used in the experiment were sown on March 20, 2021. The sampling for this study began at 8:00 on April 28, 2021.

What were the growing conditions for the plants, temperature, humidity? Which tier of leaves were used?

Line 233-234. Figure 2
The figures are too small, the signatures are illegible.

Line 324, Figure 3 – The same

Line 343, Figure 4 – The same

Do you plan to study the influence of certain bacteria on the development of rust? Has the microbial community in juniper been studied?

Author Response

Dear Reviewer,

Thank you for your time and effort on our manuscript entitled “Time-course responses of apple leaf endophytes to the infection of Gymnosporangium yamadae” (Manuscript ID: jof-2819186). All comments are valuable and very helpful for improving our manuscript. We considered all the comments carefully and highlighted all the changes within the revised manuscript by red Word. In addition, we modified the other parts of the manuscript with the red words. The point-by-point responses to the issues are listed below.

Response: R

Comments 1: What were the growing conditions for the plants, temperature, humidity? Which tier of leaves were used?

R : As described in the Materials and Methods section (located on line 106 of the manuscript), our entire experiment was conducted in a meticulously maintained clean experimental nursery, without controlling temperature and humidity. However, the timing of our inoculation coincides with the development of widespread rust diseases in the natural environment, which means that the temperature and humidity during this period are conducive to the occurrence of rust diseases. Besides, our experiments include the blank control group of healthy samples, compared with diseased group, ensuring the persuasiveness of the results. Throughout the experimental process, we selected a batch of leaf samples that sprouted at the same time and had similar growth status, thereby avoiding the influence of leaf age differences on the experimental results.

Comments 2: The figures are too small, the signatures are illegible.

R : Done. We have adjusted the font size of the captions, the dimension of the figures, and split Figure 2 into two separate figures, please see line 124, line 239, line 308, line 332 and line 350.

Comments 3: Do you plan to study the influence of certain bacteria on the development of rust? Has the microbial community in juniper been studied?

R : That would be an interesting research point, and a good innovation point for this experiment to continue, we are also paying attention and planning to do so in the next step. At present, we have not studied the microbial community in juniper, and it may become our research direction in the future.

Thank you again for your positive and constructive comments and suggestions on our manuscript.

Reviewer 4 Report

Comments and Suggestions for Authors

The manuscript presents some interesting information on the response of bacterial and fungal endophytes in apple leaves to pathogen attack which helps to better understand the complex pathosystem. Our understanding of the microbial interactions in diseased plant tissues is still very preliminary, and microbiome analyses such as the presented one are exciting new ways to get a better understanding of the dynamic microbial interactions. However, due to the limited data available, I would recommend to be very cautious with the interpretation of the results.

One of the main results of the study is the change in bacterial alpha-diversity in diseased leaves between the different stages. In line 254 the authors state: ‘the number of ASVs in the disease group dramatically increased…’ In the discussion, no persuasive explanation is given for this result. It can be assumed, that the change in alpha-diversity did not occur through ‘new’ bacterial taxa which infected the plant leaves between the different stages, but by the abundance of resident endophytic bacterial taxa. Most probably, during some stages the number of some taxa was very low, and therefore the taxa not detected. During other stages, the abundance of bacterial taxa increased and they were identified using sequencing. The caption of Figure 2 takes into consideration the ‘changes in relative abundance of endophytic bacterial and fungal communities…’, but it is not further elaborated in the result or discussion part. I think, this topic is crucial when defining and explaining microbial biodiversity and changes in alpha diversity over time.

The authors show, that the changes in relative abundance of fungal endophytes between the different infection stages were rather few compared to changes of the bacterial communities. However, it can assumed that the relative abundance of the pathogen Gymnosporangium yamadae increased significantly over time in the diseased plant leaves. Therefore, it would be expected to see an increase of the relative abundance of fungal genes belonging to the Basidiomycota, order Puccinales, from stage to stage. Of course, G. yamadae is not an endophyte, but a pathogen, and therefore not the main ‘target’ of this study. Still, it should be discussed why fungal communities in heavily infected leaves don’t show a shift towards the Puccinales.

I recommend publication after minor changes.

Figure 1: It is not clear, which stage is represented by the different graphs.

Line 36: add: “…, teliospores on juniper absorb water…”

Line 47: “…using (instead of “under”) regular laboratory methods”

Line 51:”… microorganisms that parasitize within… “(not “parasitizing”)

Line 95: Delete one “through this experiment”

Lines 106ff: The description of the apple material used for inoculation is not clear. You wrote: apple seedlings were sown on March 20. Do you mean they were planted? Only seeds can be sown, not plants. How old and tall were the apple seedling? You should include the growth conditions of the plants after inoculation (temperature, humidity, light vs. dark hours) which could have a significant effect on microbial diversity and abundance.

Line 196: What do you mean with “honey”?

Line 276: Actinobacteria are mentioned twice

Line 417 and 428: Curtobacterium instead of Curtobaterium

Line 420: some species of Pseudomonas are not beneficials, but plant pathogens

Line 427: Erwinia amylovora causes fire blight (it infects apples)

Line 439: enriched instead of enrich

Line 450: “…with the enlargement of the course of the disease”. Not clear.  Do you mean:…”disease development” or “symptom development”

Line 486: delete one “precisely”

Comments on the Quality of English Language

In general, the English is pretty good, but the manuscript should still be checked for some minor grammar mistakes.

Author Response

Dear Reviewer,

Thank you for your time and effort on our manuscript entitled “Time-course responses of apple leaf endophytes to the infection of Gymnosporangium yamadae” (Manuscript ID: jof-2819186). All comments are valuable and very helpful for improving our manuscript. We considered all the comments carefully and highlighted all the changes within the revised manuscript by red Word. In addition, we modified the other parts of the manuscript with the red words. The point-by-point responses to the issues are listed below.

Response: R

Comments 1: One of the main results of the study is the change in bacterial alpha-diversity in diseased leaves between the different stages. In line 254 the authors state: ‘the number of ASVs in the disease group dramatically increased…’ In the discussion, no persuasive explanation is given for this result. It can be assumed, that the change in alpha-diversity did not occur through ‘new’ bacterial taxa which infected the plant leaves between the different stages, but by the abundance of resident endophytic bacterial taxa. Most probably, during some stages the number of some taxa was very low, and therefore the taxa not detected. During other stages, the abundance of bacterial taxa increased and they were identified using sequencing. The caption of Figure 2 takes into consideration the ‘changes in relative abundance of endophytic bacterial and fungal communities…’, but it is not further elaborated in the result or discussion part. I think, this topic is crucial when defining and explaining microbial biodiversity and changes in alpha diversity over time.

R : In response to your question about explaining the changes of ASVs in disease progression, we have supplemented our discussion to provide further elucidation. Please see line 377,

……bacterial and fungal diversity initially showed subtle changes, and then rapidly in-creased after the maturation of spermogonia. This pattern was reflected in both the Shannon index and the abundance of Amplicon Sequence Variants (ASV), with a more pronounced increased in bacterial diversity during the later stages. This stage may represented a critical period of balance between rust pathogens and host immunity. The host's immune system continuously combated external rust pathogens during the early stages, resulting in minimal changes in diversity. However, as rust pathogens continue to colonize, the host’s cellular structure [34] and its immune system becomes severely compromised [35], the balance is broken, leading to a massive influx of for-eign communities into the host, thereby triggering a rapid increase in ASV diversity, definitely, this does not rule out the possibility that some of these microorganisms were "foreign reinforcements" recruited by the host to combat rust pathogens [36]. Simulta-neously, the endophytic microbial community strived to resist the invasion by rapidly proliferating, which may be responsible for the observed results……

We also added to the discussion on the topic of explaining changes in the relative abundance of flora over the course of disease. Please see line 407,

……4.2. Shift in composition of leaves endophytic microbial communities with G. yamadae infection.

Though there were differences in the relative abundance of endophytes between healthy and diseased leaves at various stages of disease development, the composition of high-abundance microorganisms was similar. For bacteria, Proteobacteria, Firmicutes, Actinobacteriota, Bacteroidota, and Acidobacteriota accounted for a high proportion, while the highly abundant taxa in the fungal community were mainly Ascomycota and Basidiomycota, which was consistent with many studies on endophytes in leaf microbi-ota [38,13,14].

Many members in Bacilli and Actinobacteria group have exerted growth promo-tion and antibacterial effect [39,40]. In this study, the two endophytic bacterial commu-nities were at lower levels in the diseased group before stage 6, which may be a strate-gy for rust fungi to facilitate their invasion. Previous studies show pathogens could preoccupy nutrients [41], secrete metabolites [42], creating an environment conducive to infection, all for competing with beneficial microorganisms within the host, and even studies showed pathogens inject effector proteins into antagonistic microorganisms as potent toxins to inhibit the growth of microbial competitors, which contribute to estab-lish microbial communities suitable for pathogens’ invasion in the hosts’ environments [43,44].  The phenomenon of a sharp increase in abundance during stage 6 may be at-tributed to the host's recruitment of external microorganisms, which serves as a coun-terattack mechanism of plant hosts [36].

Meanwhile, the Pleosporales in the fungal community also exhibited a similar trend of change, decreasing first and then increasing during the infection process of rust fungi. Pleosporalean fungi, reportedly, own significant antifungal properties [45,46]. In summary, the fluctuations in endophytes microbial communities may reflect a pathogen's invasion strategy……

Comments 2: The authors show, that the changes in relative abundance of fungal endophytes between the different infection stages were rather few compared to changes of the bacterial communities. However, it can assumed that the relative abundance of the pathogen Gymnosporangium yamadae increased significantly over time in the diseased plant leaves. Therefore, it would be expected to see an increase of the relative abundance of fungal genes belonging to the Basidiomycota, order Puccinales, from stage to stage. Of course, G. yamadae is not an endophyte, but a pathogen, and therefore not the main ‘target’ of this study. Still, it should be discussed why fungal communities in heavily infected leaves don’t show a shift towards the Puccinales.

R : In this study, we focused on endophytic bacteria and fungi, so the sequences of rust fungi were excluded during sequencing, we have supplemented the explanation in the manuscript, please see line 168.

……excluding the sequences of G. yamadae that we inoculated……

 Therefore, it is possible that the remaining endophytes’ sequences showed no significant changes.

Comments 3: Figure 1: It is not clear, which stage is represented by the different graphs.

R : Done. The meanings of stages 1 to 9 are mentioned in the materials and methods section, and we have also indicated them in the manuscript on line 114,

……, samples were taken at nine different time points (stage1–9) (measured by the time of inoculation)……

Comments 4: Line 36: add: “…, teliospores on juniper absorb water…”

R : Done. Please see the line 36.

……When spring rain arrives, teliospores on juniper absorb water and generate basidio-spores……

Comments 5: Line 47: “…using (instead of “under”) regular laboratory methods”

R : Done. Please see the line 47.

……, cannot be cultivated using regular laboratory methods [9]……

Comments 6: Line 95: Delete one “through’’ this experiment”

R : Done. Please see the line 96.

……Through this experiment, we aim to investigate the following……

Comments 7: Lines 106ff: The description of the apple material used for inoculation is not clear. You wrote: apple seedlings were sown on March 20. Do you mean they were planted? Only seeds can be sown, not plants. How old and tall were the apple seedling? You should include the growth conditions of the plants after inoculation (temperature, humidity, light vs. dark hours) which could have a significant effect on microbial diversity and abundance.

R : Done. Please see line 108.

……Beijing (40°0′31″N, 116°20′26″E). Three-year-old and one-meter-tall Gala apple seedlings used in the experiment were planted on March 20, 2021. The sampling for this study……

As described in the Materials and Methods section (located on line 106 of the manuscript), our entire experiment was conducted in a meticulously maintained clean experimental nursery, without controlling temperature, humidity, and light vs. dark hours. However, the timing of our inoculation coincides with the development of widespread rust diseases in the natural environment, which means that the temperature,  humidity and light vs. dark hours during this period are conducive to the occurrence of rust diseases. Besides, our experiments include the blank control group of healthy samples, compared with diseased group, ensuring the persuasiveness of the results.

Comments 8: Line 196: What do you mean with “honey”?

R : Done. This word refers to a medium for the spread of spores produced by rust bacteria when they develop into pycniopores, using “nectar” is better, and we have modified it in the manuscript.Please see line 198.

……on the spots at 14 d. Around 20 d, large drops of nectar are produced around the orange spots, aecia emerge from swollen, thickened plant tissues at 70 d. Finally, at 90 d……

Comments 9: Line 276: Actinobacteria are mentioned twice

R : Done. We have corrected the second Actinobacteria into Bacteroidota (Line 275).

……while Firmicutes, Actinobacteria and Bacteroidota decreased (Figure 3a). At the genus level……

Comments 10: Line 417 and 428: Curtobacterium instead of Curtobaterium

R :Done. Please see line 459 and 470.

……including Erwinia, Curtobacterium, Pantoea and Enterococcus……

……causes fire blight (it infects apples) [56]. Certain species in the genus Curtobacterium can cause wilt in plants [57]. Enterococcus is a……

Comments 11: Line 420: some species of Pseudomonas are not beneficials, but plant pathogens

R : The Pseudomonas group is very large, indeed encompassing plant pathogens, but it also includes many beneficial bacteria that promote plant growth and resist plant pathogens. The manuscript carefully chooses its words and utilizes "some," which is a reasonable description.Please see line 461.

……enrichment of 21 genera, with the some being beneficial bacteria such as Pseudomonas, Bacillus, Lactobacillus and Flavobacteria. While endophytes do……

Comments 12: Line 427: Erwinia amylovora causes fire blight (it infects apples)

R : Done. Please see line 469.

……Erwinia amylovora causes fire blight (it infects apples) [56].…..

Comments 13: Line 439: enriched instead of enrich

R : Done. Please see line 481).

……In addition, other enriched genera, such as Geobacillus [54]……

Comments 15: Line 450: “…with the enlargement of the course of the disease”. Not clear.  Do you mean:…”disease development” or “symptom development”

R : The continuous changes in the symptoms of rust disease occur in a sequential order, corresponding to different symptoms at different stages of the disease progression. The phrase "with the enlargement of the course of the disease" used in the manuscript is synonymous with "disease development" or "symptom development".

Comments 16: Line 486: delete one “precisely”

R : Done. We removed the first "precisely" in the revised manuscript (Line 529).

……These stages coincide precisely with the periods when rust fungi……

Thank you again for your positive and constructive comments and suggestions on our manuscript.